# The Longitudinal Relationship Between Close Friendship and Subjective Well-Being: The Chain Mediation Model of Interpersonal Trust and Perceived Social Support

**DOI:** 10.3390/bs15040480

**Published:** 2025-04-07

**Authors:** Runqing Li, Wenhu Xu, Hongyu Nie, Weida Zhang

**Affiliations:** 1School of Philosophy and Social Development, Shandong University, No. 27 Shandanan Road, Licheng District, Jinan 250100, China; 202210032@mail.sdu.edu.cn; 2Institute of Population Research, Peking University, No. 5 Yiheyuan Road, Haidian District, Beijing 100084, China; 2401111475@stu.pku.edu.cn (W.X.); hynie2020@nsd.pku.edu.cn (H.N.); 3School of Education, Jiangxi Normal University, No. 99 Ziyang Avenue, Nanchang 330224, China

**Keywords:** number of close friends, subjective well-being, perceived social support, interpersonal trust, Chinese adolescents

## Abstract

Adolescence is a critical period for developing interpersonal relationships and plays a significant role in the growth of subjective well-being. Establishing positive friendships is one of the most important predictors of subjective well-being. This study employs a longitudinal method, tracking Chinese adolescents to investigate the impact of individuals’ number of close friends on subjective well-being by examining the chain mediating roles of interpersonal trust and perceived social support. Data were collected from 987 junior high school students across five schools in Shandong Province and analysed using SPSS 27.0. The results indicate that the number of close friends at Time 1 (T1) significantly positively affects the level of subjective well-being at Time 3 (T3). At Time 2 (T2), perceived social support mediates the relationship between the number of close friends at T1 and subjective well-being at T3. Furthermore, the number of close friends at T1 has a longitudinal mediating effect on subjective well-being at T3. This effect is mediated by interpersonal trust and perceived social support at T2. This study reveals the mechanisms by which the number of close friends influences subjective well-being among Chinese adolescents. The findings highlight the significance of fostering healthy interpersonal relationships among adolescents in China. This can be achieved by promoting initiatives that enhance levels of interpersonal trust and perceived social support within communities, thereby improving overall subjective well-being among adolescents

## 1. Introduction

The desire and pursuit of well-being is a fundamental aspect of human nature ([15]). Subjective well-being (SWB) refers to an individual’s overall evaluation of their life ([7]) and serves as an important indicator of mental health in adolescents. Given that SWB is a broad and multidimensional construct, researchers have examined the different dimensions of SWB in various contexts ([70]). One of the most influential researchers in the field of SWB, [23] ([23]) posits that SWB encompasses individuals’ judgments and evaluations of their lives, shaped by both emotional experiences (comprising independent dimensions of positive and negative emotions) and cognitive aspects of life satisfaction. Currently, many researchers in psychology have adopted [26]’s ([26]) definitions and structural criteria for SWB ([13]; [26]; [51]). Essentially, SWB requires a subjective evaluation and is also a multidimensional structure ([41]). Based on contemporary academic literature, this study utilises [26]’s ([26]) framework to measure SWB, which has been empirically validated in studies involving Chinese adolescents ([67]). With the rise of positive psychology, an increasing number of researchers have embarked on a comprehensive exploration of SWB ([14]; [36]). Among these explorations, the definition of subjective well-being and the factors influencing SWB have drawn researchers’ attention ([49]; [79]), substantially advancing related theories. Research has demonstrated that strong interpersonal relationships have been shown to predict higher levels of SWB ([41]). Adolescence represents a pivotal phase for the development of prosocial behaviour ([17]) and is also a significant period for the cultivation of SWB ([29]). However, levels of SWB among adolescents tend to be relatively low, and establishing friendships is a key task during this developmental stage, highlighting the importance of tracking and enhancing adolescent mental health ([55]). Most studies have focused on students from developed countries, such as Europe and North America ([66]), with few investigations targeting students from Asian countries such as China ([40]). Therefore, this study aims to explore what factors influence the relationship between the number of close friends (NCF) and SWB among Chinese adolescents. Also, it provides new insights into the development of adolescent friendships and strategies for the long-term prevention of psychological issues.

### 1.1. The Impact of the NCF on SWB

The onset of adolescence triggers significant changes in children’s social relationships, both within families and among peers ([58]). During puberty, adolescents increasingly participate in school life, engage with peers more frequently, and rely less on parents ([8]). Research indicates that friendship quality predicts adolescents’ SWB ([47]). Self-determination theory highlights that fulfilling basic psychological needs is crucial for mental health and SWB, akin to essential physical needs ([62]). Basic psychological needs encompass three types, among which relational needs represent the concretization of the “relatedness” dimension within basic psychological needs. This concept emphasises the sense of belonging, support, and emotional connection that individuals experience in social interactions, such as in the context of friendships. Research by [18] ([18]) also demonstrates a causal relationship between relational needs satisfaction and subjective well-being. Cross-cultural research reveals that cultural backgrounds influence perceptions of relatedness, with individualistic cultures tending to see relationships as loosely connected, while collectivist cultures, like in China, emphasise social relationships’ impact on well-being ([68]). Thus, Hypothesis 1 posits that the NCF among Chinese adolescents positively predicts their SWB.

### 1.2. The Mediating Effect of Interpersonal Trust in the Relationship Between the NCF and SWB

As the Chinese proverb states, “A friend is valuable when he is well acquainted”, suggesting that individuals with more high-quality friendships benefit significantly from these relationships. Interpersonal trust (IT) refers to an individual’s generalised expectation regarding the reliability of their social partners’ statements and actions ([61]). Adolescents with friends typically exhibit higher levels of self-esteem ([1]; [45]). Moreover, individuals with high self-esteem are more likely to develop greater trust in others ([2]), and self-esteem is significantly correlated with enhanced subjective well-being (SWB) ([69]). Therefore, interpersonal trust may play a positive role in promoting subjective well-being. [31] ([31]) argue that various forms of social trust enhance well-being in both individualistic and collectivistic societies. Adolescents who trust others typically report higher SWB and life satisfaction ([60]). Hence, Hypothesis 2 proposes that IT mediates the relationship between the NCF and SWB among Chinese adolescents.

### 1.3. The Mediating Effect of Perceived Social Support in the Relationship Between the NCF and SWB

Perceived social support (PSS) refers to the emotional experience and level of satisfaction characterised by the feelings of being respected, supported, and understood within social interactions ([32]). Compared to objective social support, perceived social support is more indicative of a positive impact on mental health ([63]). During adolescence, the perception of parental support typically declines, while the perception of social support from friends tends to increase ([30]). Research indicates that the positive influence of close friendships on SWB may be associated with the social support provided by friends ([43]). The buffering hypothesis proposes that both perceived and objective social support can mitigate the negative impacts of stressful events on individuals ([44]). Friends can provide emotional support and compassion, as well as instrumental assistance during times of need ([25]). For mid-adolescents in China, support from friends is significantly correlated with the positive emotional dimension of SWB among students ([66]). Therefore, Hypothesis 3 proposes that PSS mediates the relationship between friendship networks and SWB among adolescents.

### 1.4. The Chain Mediating Effects of IT and PSS on the Relationship Between the NCF and SWB

IT is an important part of social support and psychological resources ([22]). Lower PSS is associated with lower friendship satisfaction ([73]). Building stable social relationships and experiencing positive social interactions are considered key predictors of well-being ([4]). [35] ([35]) analysed aggregated data from the Gallup World Poll conducted between 2017 and 2019 and found that six factors—income, healthy life expectancy, social support, freedom, trust, and generosity—are important predictors of overall SWB. Furthermore, research has demonstrated a strong correlation between IT and social support ([53]). Consequently, IT and PSS may serve as two chain factors influencing the relationship between the NCF and SWB among Chinese adolescents. Additionally, existing research has predominantly been conducted within developed countries, while studies on this topic in China are relatively scarce ([5]). Therefore, Hypothesis 4 proposes that IT and PSS mediate the relationship between the NCF and SWB in a chain manner.

The proposed hypothetical model is illustrated in Figure 1.

## 2. Methods

### 2.1. Participants and Procedure

The data for this study were collected from surveys conducted in Shandong Province between June 2021 and December 2022. These surveys used a multi-stage cluster sampling method focusing on students aged 12–14 (early puberty) and 15–17 (mid-adolescence), as peer relationships become more significant during mid-adolescence ([54]). All class members participated in the assessments, conducted by trained master’s or doctoral psychology students. Informed consent was obtained, allowing participants to withdraw at any time without explanation. The sample included students from five junior high schools in Jinan, Shandong Province, with 21 classes selected for a tracking survey over three time points at six-month intervals: the end of the seventh grade (T1), the end of the eighth-grade first semester (T2), and the end of the eighth-grade second semester (T3). The participant counts were 1379 at T1, 1002 at T2, and 1085 at T3. After excluding absentees and those who transferred, 987 valid samples remained from students who participated at all three time points.

### 2.2. Measures

#### 2.2.1. Number of Close Friends

The independent variable in this study was the number of friends reported by the participants. During the survey, participants were asked to indicate, “Please specify how many close friends you currently have” ([65]). “Close friends” referred to high-quality interpersonal relationships characterised by emotional depth, trust, and support ([3]), and were defined by emotional intimacy, mutual dependence, and long-term stability ([33]). In Chinese culture, they also emphasise social support and emotional reliance.

#### 2.2.2. Interpersonal Trust

The scale of interpersonal trust was based on the original version developed by [61] ([61]) and subsequently revised by [78] ([78]). The scale consisted of 6 items and used a 5-point Likert scale, with higher scores indicating a higher level of interpersonal trust in individuals. Reliability analysis was conducted using SPSS 27.0 software to calculate Cronbach’s α coefficient to assess the internal consistency of the scale. The scale demonstrated good internal consistency in this study (α = 0.81).

#### 2.2.3. Perceived Social Support

The Perceived Social Support Scale was developed by [19] ([19]) and translated and modified by Jiang Qianjin ([12]). The scale consists of 12 items, divided into three dimensions: family support, friend support, and other support. The scale uses a 5-point Likert scale ranging from “strongly disagree” to “strongly agree” (1–5). The total score for each dimension was calculated to represent the level of social support from specific sources, with higher scores indicating better perceived social support levels ([48]). In this study, the internal consistency of the PSS Scale indicated good reliability (α = 0.94).

#### 2.2.4. Subjective Well-Being

SWB is a holistic concept. In this study, the measurement of subjective well-being (SWB) included three components: positive affect, negative affect, and satisfaction with life. [24] ([24]) developed the International University Survey Scale, and Yan revised and verified it based on a Chinese background ([75]). This scale employs a 5-point Likert scoring system. Current research has only focused on the influence of friends on life satisfaction ([9]) or emotional indicators ([59]). Therefore, simultaneously considering the three dimensions of subjective well-being—life satisfaction, positive affect, and negative affect—could contribute to enriching academic research on well-being.

#### 2.2.5. Demographic Information

Based on previous research, this study considered various demographic factors that may influence SWB in developing countries ([39]; [46]; [27]; [81]). It included student ID numbers, gender, age, area (rural or urban), residence type, whether they were only children (1 = only child, 2 = non-only child), and academic performance.

### 2.3. Statistical Analysis

In this study, descriptive statistics were analysed using SPSS 27.0. Additionally, mediation effects were examined using the PROCESS 4.2 macro plugin in SPSS 27.0. PROCESS 4.2, based on regression analysis and the Bootstrap method, is capable of effectively testing multiple mediation effects and their confidence intervals ([34]). We investigated whether IT and PSS mediate the relationship between adolescent friendship networks and SWB within a chain mediation model. Specifically, PROCESS 4.2 constructs multiple regression models to estimate direct and indirect effects separately and uses the Bootstrap method (with 5000 resamples) to calculate the 95% confidence intervals for the mediation effects, thereby verifying their statistical significance ([42]).

## 3. Results

### 3.1. Common Method Bias Assessment

Harman’s single-factor test was employed to assess the common method bias across the three rounds of questionnaire administration ([83]). A common method bias is considered significant if the variance explained by the first factor is less than the critical threshold of 40%. In this study, no significant common method bias is detected.

### 3.2. Descriptive Statistics

The demographic characteristics of the participants are presented in Table 1.

Table 2 presents the descriptive statistics and correlations among the variables. The descriptive statistics reveal that the average NCF reported by the participants is 3.69 (M = 3.690, SD = 0.898). Additionally, participants reported relatively high levels of SWB (M = 4.195, SD = 1.748) and PSS (M = 3.710, SD = 0.884) while indicating relatively low levels of IT (M = 3.160, SD = 0.870). The correlational analysis demonstrates significant relationships among the variables. These correlations suggest that a greater NCF is associated with higher IT, higher PSS, and higher SWB.

### 3.3. Mediation Effect Analysis

After controlling factors such as gender, age, type of residence, and whether the participants are only children, NCF in T1 is treated as the independent variable. In contrast, the SWB from T3 serves as the dependent variable. PSS and IT from T2 are utilised as mediator variables. The study employs Model 6 from the PROCESS plugin to test the serial mediation effects. The results of this multiple mediation model are reported in Table 3, which is illustrated in Figure 2.

The results indicate that the NCF has a significant predictive effect on SWB (β = 0.597, *p* < 0.001), supporting Hypothesis 1. When PSS and IT are included in the model, the NCF significantly and positively predicts PSS (β = 0.346, *p* < 0.001) and IT (β = 0.334, *p* < 0.001). Furthermore, IT has a significant positive predictive effect on SWB (β = 0.204, *p* < 0.001), and PSS significantly and positively predicts SWB (β = 0.650, *p* < 0.001). Thus, it can be concluded that both IT and PSS serve as mediators in the relationship between the NCF and SWB. To assess the specific mediating effects, the Bootstrap method was used with 5000 samples. Table 4 presents the direct impact and the model’s mediating effect.

## 4. Discussion

This study aims to explore the relationship between NCF among adolescents and their SWB, and to examine the mediating roles of IT and PSS in this relationship. By constructing a chain mediation model, the research reveals the complex mechanisms through which the number of close friends influences subjective well-being via interpersonal trust and perceived social support. The following discussion will elaborate on these findings in detail, integrating them with the existing literature.

First, the greater the NCF among adolescents, the higher their levels of SWB, thereby supporting Hypothesis 1. This finding is consistent with previous research. Friendship is considered a significant developmental task during adolescence ([56]) and plays a crucial role in enhancing SWB. Positive social relationships can improve adolescents’ SWB ([52]). Compared to elementary school students, those in middle and high school are more likely to seek help from peers rather than teachers or parents when faced with bullying or other difficulties ([21]; [82]). Moreover, compared to “superficial friends”, close friends provide more psychological support ([77]). Close friends have a profound and lasting impact on an individual’s psychological and emotional well-being ([57]), making the establishment of intimate relationships among peers crucial ([74]). Traditional Chinese culture emphasises the principle of relationship-based connections ([11]), indicating that the development of friendships significantly impacts SWB.

Second, IT serves as a mediating factor between the NCF and SWB, aligning with Hypothesis 2. This finding reveals its specific manifestations within a cross-cultural context. Within the context of Western culture, the study by [64] ([64]) indicates a correlation between high levels of neuroticism and lower numbers of friends and IT, further emphasising the influence of personality traits in shaping social networks. Meanwhile, the work of [38] ([38]) has clearly demonstrated the importance of social capital, particularly interpersonal trust, in enhancing well-being, a conclusion that has been widely validated in Western societies. In China, we found that, although interpersonal trust still plays a positive role in promoting subjective well-being, its pathways and intensity of influence appear to exhibit unique Chinese characteristics. The application of social capital theory in the Chinese context, especially the number of close friends as a crucial component of social capital, indicates not only an individual’s deep embedding within a complex social network but also the potential for richer resource acquisition. However, a notable finding of this study, compared to Western research, is that among Chinese adolescents, while interpersonal trust serves as a mediating factor, its effect is relatively small compared to that of perceived social support. This difference may stem from the uniqueness and significance of “guanxi” (relationships) in Chinese culture, where directly perceived support and assistance resonate more strongly with individuals’ sense of well-being than mere trust.

Third, these findings not only align with the long-standing observations in Western societies that the NCF can significantly positively predict levels of SWB through social support ([72]), but they also reveal the unique expression of this relationship within a Chinese context. Particularly, the construction and maintenance of close friendships exhibit distinctive characteristics and values, thereby deepening the understanding of this universal social–psychological phenomenon. PSS mediates the relationship between the NCF and SWB. Numerous studies have found a close relationship between social support and friendship ([71]), suggesting that friends provide support, fulfil emotional needs, and offer behavioural feedback ([50]). According to self-determination theory ([20]), adolescents who receive social support typically exhibit better physical and mental health and fewer behavioural problems ([37]), thereby contributing more effectively to the enhancement of subjective well-being. In Chinese society, the role of close friends often transcends that of mere emotional or instrumental support providers; it is more intricately intertwined with cultural elements such as “renqing” (reciprocity) and “mianzi” (social standing, dignity, and reputation), creating a more complex and profound social exchange network ([80]). This unique cultural phenomenon makes Chinese individuals place greater emphasis on the closeness, trustworthiness, and long-term reciprocity of relationships when seeking and providing social support, thereby strengthening the role of social support in promoting subjective well-being.

Fourth, this study reveals the chain mediating effects of IT and PSS in the relationship between the NCF and SWB. It underscores the complex mechanisms underlying adolescents’ socio-psychological development. In Western culture, [10] ([10]) highlights the critical role of IT in promoting social support, noting that high levels of trust encourage adolescents to explore and accept help from others proactively. This aligns with the results of the study, further validating trust’s mediating role in building positive social relationships. At the same time, this study emphasises the distinctiveness of social relationships within the context of Chinese culture. Chinese culture emphasises “renqing” (reciprocity) and “guanxi” (relationship). In this cultural context, interpersonal trust is not merely a prerequisite for the transmission of social support but also a core element in maintaining harmonious relationships and facilitating resource sharing and emotional exchange ([76]). Thus, for Chinese adolescents, an increase in their NCF signifies not just an expansion of their social network but also deeper emotional connections and social resources. Additionally, this study echoes the main effect theory of social support ([16]), which posits that social support itself has a direct and significant positive impact on individuals’ mental health and well-being. In Chinese society, frequent and positive interpersonal interactions are regarded as significant indicators of personal value and social status. Such interactions not only fulfil adolescents’ needs for belonging and self-esteem but also indirectly enhance their subjective well-being by strengthening their sense of social identity and self-efficacy. Although [6]’s ([6]) research did not specifically target Chinese adolescents, its discussion of the relationship between positive interactions and well-being has been cross-culturally validated in this study and has demonstrated even richer implications within the Chinese cultural context.

There are some limitations in this research. Firstly, the self-report questionnaires used in the study may introduce biases in the results. For instance, some adolescents may, due to narcissism, believe they have many close friends, leading them to perceive themselves as favoured adolescents ([28]). Secondly, the study’s sample was limited to second- and third-year junior high school students. Future research could expand the age range of participants to verify the findings in a broader population across various groups.

## 5. Conclusions

This study illuminates that the NCF significantly positively predicts the level of SWB. Furthermore, both IT and PSS mediate jointly in this relationship. For practical implications, the dynamic connection between close friendships and SWB, along with the complex interactive patterns formed by IT and PSS, constitute an important foundation for promoting adolescents’ mental health and well-being. Adolescents should seek effective support and resources to enhance their mental health and sense of well-being. Our specific recommendations are as follows: Adolescents should build and strengthen their social support networks, particularly close friendships, by understanding their social needs and participating in activities that promote emotional exchange, along with receiving personalised psychological counselling. They should focus on establishing and maintaining trust within these friendships by creating targeted interactive groups that teach communication skills, emotional intelligence, and conflict resolution, facilitating effective communication for trust repair when necessary. Additionally, adolescents can enhance their awareness of social support by reflecting on their networks, expressing gratitude, and sharing experiences, which will help them proactively seek and utilise social support resources.

## Figures and Tables

**Figure 1 behavsci-15-00480-f001:**
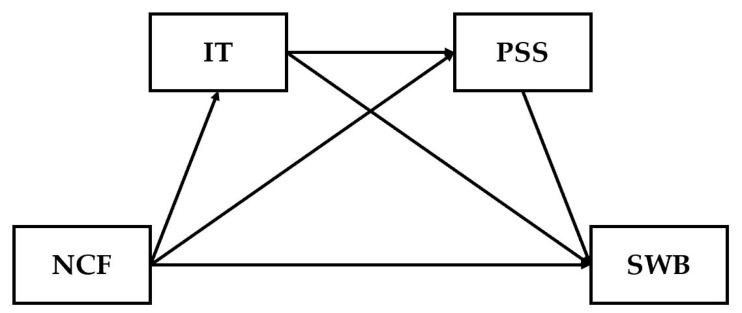
Research Model. Note: The abbreviations NCF, IT, PSS, and SWB appearing in Figure 1 represent the number of close friends, interpersonal trust, perceived social support, and subjective well-being, respectively.

**Figure 2 behavsci-15-00480-f002:**
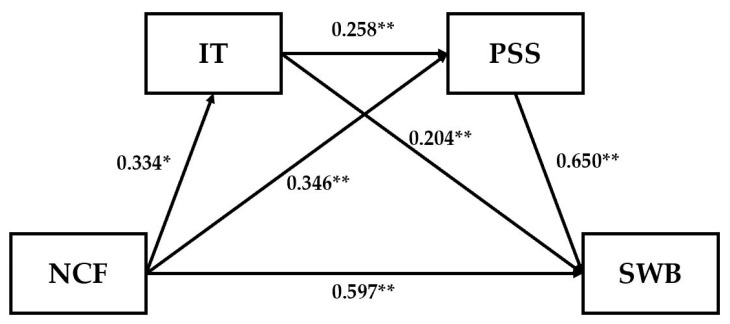
The path coefficient of the chain mediation model. * *p* < 0.05, ** *p* < 0.01.

**Table 1 behavsci-15-00480-t001:** Demographic information.

Variable	Options	Percentage	Number of Responses
Gender	Female	42	415
	Male	58	572
Age	13	5.4	53
	14	58.6	578
	15	31.3	309
	16	4.7	47
Residence type	Urban	41.3	408
	Rural	58.7	579
Only child	Yes	44.3	437
	No	55.7	550
Achievement	Below Expectations	13.1	129
	Below Average	25.4	251
	Average	34.2	338
	Above Average	19.9	196
	Good	7.4	73

**Table 2 behavsci-15-00480-t002:** Descriptive statistics and correlational analysis.

Variable	M	SD	T1 NCF	T2 IT	T2PSS	T3 SWB
T1 NCF	3.690	0.898	1			
T2 IT	3.160	0.870	0.337 **	1		
T2 PSS	3.710	0.884	0.423 **	0.362 **	1	
T3SWB	4.195	1.748	0.285 **	0.260 **	0.428 **	1

** *p* < 0.01.

**Table 3 behavsci-15-00480-t003:** Regression analysis.

DV	IV	R	R2	F	β	*p*
T3SWB	T1NCF	0.349	0.122	22.618	0.597	0.000
T2IT	T1NCF	0.349	0.122	22.655	0.334	0.000
T2PSS	T1NCF	0.511	0.261	49.386	0.346	0.000
	T2IT				0.258	0.000
T3SWB	T1NCF	0.484	0.234	37.404	0.248	0.000
	T2IT				0.204	0.001
	T2PSS				0.650	0.000

**Table 4 behavsci-15-00480-t004:** The chain mediating effect path.

Effect	Path	SE	Proportion	95% Confidence Interval	
				LLCI	ULCI
Direct effect	T1 NCF → T3 SWB	0.248	41.54%	0.127	0.369
Indirect effect 1	T1 NCF → T2 IT → T3 SWB	0.068	11.39%	0.028	0.115
Indirect effect 2	T1 NCF → T2 PSS → T3 SWB	0.225	37.69%	0.167	0.288
Indirect effect 3	T1 NCF → T2 IT → T2 PSS → T3 SWB	0.056	9.38%	0.035	0.082
Total indirect effect		0.349	58.46%	0.278	0.424
Total effect	T1 NCF → T3 SWB	0.597		0.483	0.711

## Data Availability

The data presented in this study are available on request from the corresponding author.

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
