# Peer review of "The Longitudinal Relationship Between Close Friendship and Subjective Well-Being: The Chain Mediation Model of Interpersonal Trust and Perceived Social Support"

_behavsci, 2025, doi:10.3390/bs15040480_

Round 1
Reviewer 1 Report
Comments and Suggestions for Authors
Before I start, I would like to congratulate you on the high number of participants. However, I have noticed some important shortcomings that need to be addressed. First of all you abbreviate number of close friends to (NCF) (page 2 line 63). But on the same page, line 82, you abbreviate needs of comradeship and friendship (NCF). This influences the whole manuscript as it is not clear exactly what you are referring to when you name NCF. I suggest, abbreviate close friendships as (CF).
In addition, previously abbreviated words are abbreviated again, for example on page 3, lines 73 and 94. I recommend revising the whole manuscript to correct the abbreviations.
On the other hand, on a general level, I recommend that you improve the drafting of the text by linking the ideas better. For example, on page 3, lines 85-90. They do not have an argumentative nexus, they seem more like single sentences. Revise the whole text.
On page 3 lines 73-77 they link the theory of basic psychological needs with the article by cuijpers et al (2022) when the latter in his article does not mention anything about it. So they are not talking about the same needs. They should clarify whether they are referring to Decy and Ryan's theory or to something else. As well as citing where Vitamin ‘S’ comes from.
I suggest changing the aim of the study to: "Therefore, this study aims to explore what factors influence the relationship between the number of close friends (NCF) and SWB among Chinese adolescents". Since you are looking for what elements influence within that relationship.
How and when were the instruments supplied? Reference should be made to this question to further clarify the data collection procedure.
Figure 1 and 2 would show correlations I understand, so they would be misrepresented. Since there is no causality but a relationship that can be inverse, so the arrows should go in both directions.
Add quote on page 4 line 14.
You name a number of instruments but do not indicate how they work, nor how the data have been treated, as you only indicate an overall average value per factor. I would recommend further explanation of each instrument and describe how each instrument has been processed to arrive at that mean value.
Table 1 shows a leftward shift in the latest data. On the other hand, they incorporate the (statistical) value, in my opinion it is not necessary and I would prefer to see the number of answers next to the percentage.
Point 2.3 should be data analysis or statistical analysis. They also talk about the PROCESS 4.2 plugin but do not explain exactly how it works.
Analysing table 3 and figure 2 I observe discrepancies in factor IV. Check the data and the abbreviations as figure 2 relates NCF to PSS 0.346 but this does not appear in table 3. I also suggest reviewing all tables and incorporating the explanation of the abbreviations as a footnote to the table.
As far as the discussion is concerned, it is first important to go back to the objective of the study and then to discuss the order of the results. Moreover, it is rather brief. I recommend adding the section to discuss the limitations of the study and future proposals in the discussion and not in the conclusions in order to give it more depth. Correct Line 24: third
In the references section check for missing elements such as DOIs.
Author Response
Comments1:[First of all you abbreviate number of close friends to (NCF) (page 2 line 63). But on the same page, line 82, you abbreviate needs of comradeship and friendship (NCF). This influences the whole manuscript as it is not clear exactly what you are referring to when you name NCF. I suggest, abbreviate close friendships as (CF).In addition, previously abbreviated words are abbreviated again, for example on page 3, lines 73 and 94. I recommend revising the whole manuscript to correct the abbreviations]
Response1:[Abbreviation consistency: To ensure clarity, we have removed the abbreviation “needs of comradeship and friendship (NCF)” and standardized “NCF” to exclusively refer to “number of close friends” throughout the manuscript, including in the framework diagram. Additionally, we have conducted a thorough review to ensure that all abbreviations are properly defined upon first use and remain consistent throughout the text.]
Comments2:[On the other hand, on a general level, I recommend that you improve the drafting of the text by linking the ideas better. For example, on page 3, lines 85-90. They do not have an argumentative nexus, they seem more like single sentences. Revise the whole text.]
Response2:[Refinement of logical coherence: We have carefully reviewed the manuscript, particularly lines 85-90 on page 3, where we refined the logical flow between sentences. Consequently, we have removed references that were no longer necessary due to these modifications.]
Comments3:[On page 3 lines 73-77 they link the theory of basic psychological needs with the article by cuijpers et al (2022) when the latter in his article does not mention anything about it. So they are not talking about the same needs. They should clarify whether they are referring to Decy and Ryan's theory or to something else. As well as citing where Vitamin ‘S’ comes from.]
Response3:[Theoretical clarification: We have elaborated on the relationship between Deci and Ryan’s Basic Psychological Needs Theory and the concept of relatedness needs as discussed by Cuijpers et al. (2022), refining the logical connections in this section.]
Comments4:[I suggest changing the aim of the study to: "Therefore, this study aims to explore what factors influence the relationship between the number of close friends (NCF) and SWB among Chinese adolescents". Since you are looking for what elements influence within that relationship.]
Response4:[Modification of research objectives: We have revised the research objectives based on your suggestions to enhance clarity and alignment with the study’s focus.]
Comments5:[How and when were the instruments supplied? Reference should be made to this question to further clarify the data collection procedure.]
Response5:[Details on data collection: We have supplemented the methodology section with additional details regarding the specific timeframe and procedures of data collection to improve transparency and reproducibility.]
Comments6:[Figure 1 and 2 would show correlations I understand, so they would be misrepresented. Since there is no causality but a relationship that can be inverse, so the arrows should go in both directions.]
Response6:[Regarding the direction of arrows in Figures 1 and 2: We sincerely appreciate your input on this matter. After reviewing relevant literature on mediation effects, we found that single-headed arrows are the conventional representation in similar studies. Therefore, we have retained this format.]
Comments7:[Add quote on page 4 line 14]
Response7:[We may not have fully understood your suggestion regarding the additional citation. If you could provide further clarification, we would be grateful. We want to ensure that all references are appropriately incorporated to enhance the manuscript’s rigor.]
Comments8:[You name a number of instruments but do not indicate how they work, nor how the data have been treated, as you only indicate an overall average value per factor. I would recommend further explanation of each instrument and describe how each instrument has been processed to arrive at that mean value.]
Response8:[We may not have fully understood your suggestion regarding the additional citation. If you could provide further clarification, we would be grateful. We want to ensure that all references are appropriately incorporated to enhance the manuscript’s rigor.We have provided a more detailed explanation of the principles underlying the analytical tools used in our study, as well as the data processing methods, to enhance methodological transparency.]
Comments8:[Table 1 shows a leftward shift in the latest data. On the other hand, they incorporate the (statistical) value, in my opinion it is not necessary and I would prefer to see the number of answers next to the percentage.]
Response8:[As per your suggestion, we have removed statistical values and added exact response counts next to percentage figures. Additionally, we have checked and corrected the issue of left-skewed data distribution.]
Comments9:[Point 2.3 should be data analysis or statistical analysis. They also talk about the PROCESS 4.2 plugin but do not explain exactly how it works.]
Response9:[We have changed the section title to “Statistical Analysis” and provided a more detailed explanation of the PROCESS 4.2 plugin in the text.]
Comments10:[Analysing table 3 and figure 2 I observe discrepancies in factor IV. Check the data and the abbreviations as figure 2 relates NCF to PSS 0.346 but this does not appear in table 3. I also suggest reviewing all tables and incorporating the explanation of the abbreviations as a footnote to the table.]
Response10:[Thank you for pointing this out. We have highlighted in red the correlation between “NCF” and “PSS” (0.346) in Table 3 and added detailed explanations of abbreviations below the table. Additionally, to improve clarity, we have repositioned Figure 2 to directly follow Table 2.]
Comments11:[As far as the discussion is concerned, it is first important to go back to the objective of the study and then to discuss the order of the results. Moreover, it is rather brief. I recommend adding the section to discuss the limitations of the study and future proposals in the discussion and not in the conclusions in order to give it more depth.]
Response11:[We have restructured the discussion to align more closely with the research objectives and present the findings in a logical sequence. Furthermore, we have deepened the discussion to enhance the argumentation. Additionally, we have moved the “Research Limitations and Future Directions” subsection from the conclusion to the discussion for better structural coherence. Finally, we have corrected the spelling error in “third.”]
Comments12:[Correct Line 24: third]
Response12:[We have corrected the term “third” in line 24 to ensure accuracy.]
Comments12:[In the references section check for missing elements such as DOIs.]
Response13:[We have verified and added any missing DOI information to ensure compliance with the journal’s formatting requirements. Some references, particularly older books, do not have corresponding DOIs.]
Reviewer 2 Report
Comments and Suggestions for Authors
On the first paragraph in the Introduction section, the authors state this "One of the most influential researchers in the field of SWB, Diener et al. (2018)". If they refer to one of the most influential, then only that author should be cited and not more. I recommend citing a research where only Diener appears.
On the same first paragraph, the authors make a comparison between Europe and North America, stating that China is a developing country, which is not accurate. Read this for a 2020 list of developing countries (China is not listed there and since 2020 it has certainly grew) - https://www.5-cc.com/en/2025/register-here/developing-countries/. While I understand the idea, it is best to make the comparison between Europe/North America and Asian countries perhaps?.. Just an idea...
On chapter 1.1, the authors mention this "The importance of relatedness has led researchers to label it as "Vitamin S" (Vitamin Social Contact)." I recommend citing several such researchers, as the idea is intriguing and readers might want to check on it.
At subchapter 2.2.1., the participants were asked to “... specify how many close friends you currently have”, but I could not find in the article a working definition of the concept. Have the participants been instructed about what close friends mean in the author's view? Close friends can have a lot of different meanings...
In subchapter 2.2.5., I have not seen the area (rural or urban) demographic factor. In terms of friendships, this could be an interesting factor to consider and know. Perhaps the regions the authors mentioned are clear for someone living in China, but for people outside of China, this is not clear.. (it does indeed appear in table 1, in subchapter 3.2, but could also be mentioned above, at 2.2.5)
Overall, the citations are ok. They could be a bit more updated (some of them are 30-40 years old) and in some places they don't respect the APA citing style (for example, Diener et al., 2000).
Author Response
Comments1:[On the first paragraph in the Introduction section, the authors state this "One of the most influential researchers in the field of SWB, Diener et al. (2018)". If they refer to one of the most influential, then only that author should be cited and not more. I recommend citing a research where only Diener appears. ]
Response1:[Following your recommendation, we have revised the first paragraph of the introduction, replacing the citation of “Diener et al. (2018)” with a reference to Diener’s individual work (Diener, 1984) to more accurately reflect his authoritative contributions to the field of subjective well-being (SWB).]
Comments2:[On the same first paragraph, the authors make a comparison between Europe and North America, stating that China is a developing country, which is not accurate. Read this for a 2020 list of developing countries (China is not listed there and since 2020 it has certainly grew) - https://www.5-cc.com/en/2025/register-here/developing-countries/. While I understand the idea, it is best to make the comparison between Europe/North America and Asian countries perhaps?.. Just an idea...]
Response2:[To avoid potential misunderstandings, we have modified the phrase referring to China as a developing country and instead revised it to “China and other Asian countries.”]
Comments3:[On chapter 1.1, the authors mention this "The importance of relatedness has led researchers to label it as "Vitamin S" (Vitamin Social Contact)." I recommend citing several such researchers, as the idea is intriguing and readers might want to check on it.]
Response3:[In Section 1.1, we have refined the logical structure of this part, emphasizing the number of close friends as a key aspect of adolescents’ need for relationships, given its central role in our study. Due to time constraints, we have not yet expanded on the research related to "Vitamin S," but we will incorporate additional details if needed.]
Comments4:[At subchapter 2.2.1., the participants were asked to “... specify how many close friends you currently have”, but I could not find in the article a working definition of the concept. Have the participants been instructed about what close friends mean in the author's view? Close friends can have a lot of different meanings.]
Response4:[In Section 2.2.1, we have added an operational definition of “close friends” to enhance clarity.]
Comments5:[In subchapter 2.2.5., I have not seen the area (rural or urban) demographic factor. In terms of friendships, this could be an interesting factor to consider and know. Perhaps the regions the authors mentioned are clear for someone living in China, but for people outside of China, this is not clear.. (it does indeed appear in table 1, in subchapter 3.2, but could also be mentioned above, at 2.2.5)]
Response5:[In Section 2.2.5, we have included a discussion of urban and rural demographic factors, along with relevant references to strengthen this aspect.]
Comments6:[Overall, the citations are ok. They could be a bit more updated (some of them are 30-40 years old) and in some places they don't respect the APA citing style (for example, Diener et al., 2000).]
Response6:[We have thoroughly reviewed and updated the formatting of all references to ensure consistency with the journal’s requirements. Additionally, we have prioritized using the most up-to-date sources wherever possible, except for a few authoritative classic references.]
Round 2
Reviewer 1 Report
Comments and Suggestions for Authors
There are only a few small errors to be corrected:
Page 2 line 58 and 76 the sentence is abbreviated again.
Page 2 line 89 remove one among of the sentence.
Page 3 line 130: you need to insert a reference that support your afirmation: "...as peer relationships become more significant during mid-adolescence."
Page 4 line 150 an 158 : change scoring system for Likert scale
Table 1 add: Number of responses not only ‘numbers’.
Table 3: I would say that T2PT should be T2IT.
Author Response
Comment 1: [Page 2, line 58 and 76 – The sentence is abbreviated again.]
Response1: [Thank you for pointing this out. We have revised the sentences on page 2, lines 58 and 76, to ensure they are complete and clear.]
Comment 2: [Page 2, line 89 – Remove one “among” in the sentence.]
Response2: [We have removed the redundant “among” in the sentence on page 2, line 89.]
Comment 3: [Page 3, line 130 – Insert a reference to support the statement: “...as peer relationships become more significant during mid-adolescence.”]
Response3: [We appreciate this suggestion. We have added a relevant reference to support this statement.]
Comment 4: [Page 4, lines 150 and 158 – Change the scoring system for the Likert scale.]
Response4: [Thank you for highlighting this issue. We have updated the scoring system for the Likert scale on page 4, lines 150 and 158.]
Comment 5: [Table 1 – Add “Number of responses” instead of only “numbers.”]
Response5: [We have revised Table 1 to include “Number of responses” as suggested.]
Comment 6: [Table 3 – Change “T2PT” to “T2IT.”]
Response6: [We have updated Table 3 to replace “T2PT” with “T2IT” as recommended.]
